# Clinical Implication of Preoperative Renal Function on Oncological Outcomes in Patients with Upper Tract Urothelial Carcinoma after Radical Nephroureterectomy

**DOI:** 10.3390/biomedicines10061340

**Published:** 2022-06-07

**Authors:** Tae Heon Kim, Hyun Hwan Sung, Jong Jin Oh, Seok Ho Kang, Ho Kyung Seo, Bumsik Hong, Ja Hyeon Ku, Byong Chang Jeong

**Affiliations:** 1Department of Urology, CHA Bundang Medical Center, CHA University, Seongnam 131-081, Korea; theon.kim@cha.ac.kr; 2Department of Urology, Samsung Medical Center, Sungkyunkwan University School of Medicine, Seoul 06531, Korea; hyunhwan.sung@samsung.com; 3Department of Urology, Seoul National University Bundang Hospital, Seongnam 131-081, Korea; urojj@snubh.org; 4Department of Urology, Korea University Anam Hospital, Korea University College of Medicine, Seoul 06531, Korea; mdksh@korea.ac.kr; 5Department of Urology, National Cancer Center, Goyang 10408, Korea; seohk@ncc.re.kr; 6Department of Urology, Asan Medical Center, University of Ulsan College of Medicine, Seoul 06531, Korea; bshong@amc.seoul.kr; 7Department of Urology, Seoul National University Hospital, Seoul National University College of Medicine, Seoul 06531, Korea

**Keywords:** urothelial carcinoma, upper urinary tract, nephroureterectomy, kidney function, survival

## Abstract

This study aims to evaluate the impact of preoperative renal function on oncological outcomes in patients with upper tract urothelial carcinoma (UTUC) who underwent radical nephroureterectomy (RNU). We performed a retrospective analysis of patients who underwent RNU between 2000 and 2012 at six academic centers. The patients were stratified into two groups based on preoperative renal function: eGFR < 60 mL/min/1.73 m^2^ (chronic kidney disease; CKD) and eGFR ≥ 60 mL/min/1.73 m^2^ (non-CKD). We investigated oncological outcomes, including overall survival, cancer-specific survival, and progression-free survival dichotomized by preoperative renal function. Multivariable Cox proportional hazards regression was used to determine if preoperative CKD was associated with oncological outcomes. In total, 1733 patients were eligible for the present study (CKD = 707 and non-CKD = 1026). Significant differences were noted in the clinical and pathologic features among the two groups, including age, sex, tumor localization, pathological T stage, tumor grade, and number of patients who received adjuvant chemotherapy. The estimated five-year overall survival (79.4 vs. 67.5%, log-rank *p* < 0.001), cancer-specific survival (83.5 vs. 73.6%, log-rank *p* < 0.001), and progression-free survival (74.6 vs. 61.5%, log-rank *p* < 0.001) were significantly different between the two groups, longer in the non-CKD group. Upon multivariable analysis, preoperative CKD status was associated with increased risk of overall mortality, cancer-specific mortality, and progression (*p* = 0.010, *p* = 0.016, and *p* = 0.008, respectively). UTUC patients with preoperative CKD had a higher risk of poor overall survival, cancer-specific survival, and progression-free survival after RNU than those without CKD.

## 1. Introduction

Upper tract urothelial carcinoma (UTUC) is a relatively rare malignancy that accounts for 5–10% of urothelial carcinomas overall, with an estimated annual incidence of nearly two cases per 100,000 population [1,2,3]. Radical nephroureterectomy (RNU) with bladder cuff excision is considered the mainstay of treatment for UTUC because it offers adequate local tumor control and better long-term survival. However, over the past several decades, survival outcomes following RNU have not shown improvement [2,4,5]. The prognosis of patients with UTUC remains poor, with relatively high recurrence rates > 30% at five years and estimates of cancer-specific survival ranging between 60–80% at five years [6,7]. Considering the poor prognosis of UTUCs treated with RNU, the addition of perioperative chemotherapy can be expected to have a synergistic effect to improve patients’ outcomes [8,9]. Several prognostic factors have been investigated to allow clinicians to risk stratify and better select patients for perioperative chemotherapy, either in the neoadjuvant or adjuvant setting. In previous studies of UTUC patients following RNU, several factors, such as old age, tumor stage, tumor grade, tumor size, lymphovascular invasion (LVI), and lymph node involvement, are significant predictors of poor oncological outcomes [10].

Chronic kidney disease (CKD) has long been recognized as an independent predictor of adverse postoperative outcomes including cardiopulmonary complications, prolonged length of hospital stay, and increased mortality for patients undergoing vascular, general, and urological oncological surgery [11,12]. Recent data have revealed an association between preoperative CKD in UTUC and poor prognosis [11,12,13], but no high-level evidence based on randomized trials has been published. Conclusive data regarding the impact of CKD on prognosis of patients with UTUC are lacking. Although some reports suggest that preoperative CKD is a predictive factor for oncological outcomes following RNU in patients with UTUC [13,14,15], another report demonstrated that preoperative renal function was not related to local recurrence or cancer-specific survival (CSS) [16]. Thus, no consensus exists on the prognostic role of preoperative CKD status in UTUC management.

In this context, we conducted a large, multi-institutional, retrospective study to evaluate the impact of preoperative renal insufficiency on oncological outcomes in patients who were treated with RNU for UTUC.

## 2. Materials and Methods

### 2.1. Study Design, Patient Selection, and Data Collection

We retrospectively collected data from patients with UTUC who underwent RNU between January 2000 and December 2012 at six academic centers participating in the Urothelial Cancer-Advanced Research and Treatment (UCART) study group. Institutional review board approval was obtained by each participating center as needed. The requirement for informed consent was waived by each institutional review board due to the retrospective nature of the study. This retrospective study was carried out according to the STROBE (Strengthening the Reporting of Observational Studies in Epidemiology) statement [17]. Exclusion criteria were bilateral tumor, clinical evidence of distant metastasis, history of previous UTUC or other malignancies, previous or concomitant radical cystectomy, neoadjuvant chemotherapy, and missing data on confounding variables. All patients underwent standard RNU with removal of the kidney, ureter, and ipsilateral bladder cuff. The following surgical modalities were included in the analysis: open, laparoscopic, and robot-assisted approaches. Bladder cuff excision was performed through either an intravesical or extravesical approach according to surgeon discretion. Regional lymph node dissection was performed only if enlarged lymph nodes were observed on preoperative imaging study or were detected in the operative field during surgery. Clinical and pathological data (age at surgery, sex, previous bladder cancer, tumor location, tumor size, pathological T stage, lymph node status, tumor grade, presence of LVI, adjuvant chemotherapy, follow-up time, and oncologic outcomes) were recorded in their respective institutional databases. All surgical specimens were processed according to standard pathological procedures at each institution. Tumors were staged according to the 2010 American Joint Committee on Cancer/International Union Against Cancer TNM classification [18]. Tumor grading was assessed according to the 1998 World Health Organization/International Society of Urologic Pathology consensus classification [19]. Adjuvant chemotherapy was mainly considered in patients with pT2-4 or pN+ disease, but utilization of adjuvant chemotherapy was determined by several factors, including tolerability of chemotherapy and patient’s motivation. Preoperative renal function was evaluated using the estimated glomerular filtration rate (eGFR) within 30 days before RNU. The eGFR was calculated from the serum creatinine using the Modification of Diet in Renal Disease formula, which adjusts for age and sex [20].

### 2.2. Patient Follow-Up

Due to the multi-institutional retrospective nature of this study, postoperative follow-up regimens across the participating institutions were not standardized but consisted of evaluations every three months for the first two years after RNU, every six months for the next two to three years and annually thereafter. Urine cytology, cystoscopy, and routine checkups that included history taking, blood tests, and physical examinations were conducted at each visit. Imaging evaluations using chest radiography and computed tomography scans for the abdomen and pelvis were also completed during each visit. If clinically indicated and available, bone scans and chest computed tomography were performed.

### 2.3. Definitions of Variables for Analyses and Outcomes

Based on the National Kidney Foundation guidelines, CKD was defined as eGFR < 60 mL/min/1.73 m^2^ [21]. We stratified the patients into the following two groups according to preoperative renal function: eGFR < 60 mL/min/1.73 m^2^ (CKD group) and eGFR ≥ 60 mL/min/1.73 m^2^ (non-CKD group). Oncological outcomes of this study were overall survival (OS), CSS, progression-free survival (PFS), and intravesical recurrence-free survival (IVRFS). OS and CSS were defined as the time until death from any cause and the time until any death secondary to UTUC, respectively. Progression-free survival was defined as the time to tumor relapse in the operative field or regional lymph node or distant metastases. Intravesical recurrence-free survival was defined as the time to tumor relapse in the bladder. Time to the event was calculated from the date of RNU.

### 2.4. Statistical Analysis

Descriptive statistics (including medians and interquartile ranges [IQRs]) were reported for continuous variables, while frequencies and proportions were compiled for categorical variables. Clinical and pathological characteristics were compared between groups using the Mann–Whitney U test and the chi-square tests for continuous and categorical variables, respectively. Survival curves were plotted using the Kaplan–Meier method to describe OS, CSS, PFS, and IVRFS and were compared using the log-rank test. Patients were censored if lost to follow-up or at the end of study follow-up if they did not experience the events of interest. Cox proportional hazard regression analyses were performed to investigate the association of preoperative renal insufficiency and oncological outcomes of OS, CSS, PFS, and IVRFS. All statistical tests were two-sided with a significance level set at *p* < 0.05. We calculated all statistical analyses using the Statistical Package for the Social Sciences (SPSS^®^) for Windows, version 25.0 (IBM Corporation, Armonk, NY, USA).

## 3. Results

A total of 1733 patients with non-metastatic UTUC who underwent RNU was included in the final cohort. For the entire cohort, the median patient age at RNU was 65 years (IQR: 58–72), and the median follow-up duration after RNU was 55.7 months (IQR: 31.2–89.9). Of the 1733 patients with UTUC, 707 (40.8%) were in the eGFR < 60 mL/min/1.73 m^2^ group, and 1026 (59.2%) had eGFR ≥ 60 mL/min/1.73 m^2^. The clinical and pathologic features of the cohort stratified by preoperative CKD status are shown in Table 1. Median age and proportions of patients with more advanced pathologic T stage and higher grade who were also receiving adjuvant chemotherapy were significantly higher in the CKD group than in the non-CKD group.

The median length of follow-up for patients who survived was 61.9 months (IQR: 36.9–94.0) for the CKD group and 71.2 months (IQR: 46.5–105.4) for the non-CKD group. The proportion of patients who achieved longer than five years of follow-up was 40.3% (285/707) and 50.7% (520/1026) for the CKD and non-CKD groups, respectively. The CKD group had significantly worse oncological outcomes than the non-CKD groups in terms of PFS, CSS, and OS. Overall, 476 patients died from any cause during follow-up (249 in the CKD group and 227 in the non-CKD group); 354 UTUC-related deaths occurred in 187 patients in the CKD group and 167 in the non-CKD group. The five-year OS estimate was 67.5% for those with CKD and 79.4% for those with non-CKD (*p* < 0.001, Figure 1A). The five-year CSS for CKD patients was 73.6% vs. 83.5% for non-CKD patients (*p* < 0.001, Figure 1B). In total, 513 patients progressed to tumor relapse within the operative field or distant metastasis: 258 in the CKD group and 255 in the non-CKD group. The five-year PFS estimate was 61.5% for CKD patients and 74.6% for non-CKD patients (*p* < 0.001, Figure 1C). During follow-up, there were 694 intravesical recurrences, including 281 in the CKD group and 413 in the non-CKD group. The five-year IVRFS estimates were 54.2% and 56.4% for patients treated with CKD or non-CKD, respectively, and this difference was not statistically significant (*p* = 0.211, Figure 1D).

Multivariable analyses revealed that, after adjustment for age, pathologic T stage, lymph node status, tumor grade, and receipt of adjuvant chemotherapy, preoperative CKD status imparted higher risk of death from any cause (hazard ratio [HR]: 1.30, 95% CI: 1.07–1.59; *p* = 0.010), cancer-specific mortality (HR: 1.33, 95% CI 1.05–1.67; *p* = 0.016), and progression to tumor relapse in the operative field or distant metastasis (HR: 1.31, 95% CI: 1.07–1.59; *p* = 0.008), but this trend was not significant for IVRFS (HR: 1.18, 95% CI: 1.00–1.39; *p* = 0.056; Table 2).

## 4. Discussion

Based on a large multi-institutional database of patients treated with RNU, our results confirmed that preoperative CKD status was associated with worse OS, CSS, and PFS compared to those outcomes in non-CKD patients. This study demonstrated that preoperative renal insufficiency confers poor oncological outcomes in patients with UTUC independent of typical risk factors. Our results suggest that preoperative renal function can be used as an independent prognostic factor for UTUC.

According to European Association of Urology guidelines [2], established predictors of UTUC prognosis include age, smoking, obesity, tumor location, multifocality, size, hydronephrosis, and pathological factors but do not include preoperative renal function. No clear relationship between CKD and prognosis was found in UTUC, although most recent publications have demonstrated a close relationship between preoperative renal function and adverse oncological outcomes in that preoperative CKD status is associated with higher risk of recurrence and cancer-specific mortality in UTUC [13,14,15]. Kodama et al. [14] studied 426 UTUC patients and reported the negative impact of preoperative CKD on poor visceral recurrence-free survival, CSS, and OS after RNU. Motoma et al. [15] retrospectively reviewed 456 patients with UTUC who underwent RNU and reported preoperative renal impairment (defined as eGFR < 60 mL/min/1.73 m^2^) to be associated with worse oncological outcomes than those in patients with eGFR ≥ 60 mL/min/1.73 m^2^ and highlighted that the severity of preoperative renal insufficiency was strongly associated with a lower survival probability. Indeed, eGFR < 45 mL/min/1.73 m^2^ demonstrated significantly worse oncological outcomes than 45 ≤ eGFR < 60 mL/min/1.73 m^2^ in terms of recurrence-free survival (*p* = 0.010), CSS (*p* = 0.022), and OS (*p* = 0.021). Their multivariate analysis showed that preoperative eGFR < 45 mL/min/1.73 m^2^ was an independent predictor for worse recurrence-free survival (HR: 2.9, 95% CI: 1.6–5.3; *p* < 0.001), CSS (HR: 2.8, 95% CI: 1.2–6.3; *p* < 0.001), and OS (HR: 2.2, 95% CI: 1.1–4.3; *p* < 0.001) compared with eGFR ≥ 60 mL/min/1.73 m^2^. These findings were in accordance with our results. This study included a large (n = 1733) multi-institutional series of patients with a median follow-up of 56 months (IQR: 31–90). To the best of our knowledge, the current study represents one of the largest series of UTUC patients treated with RNU in the literature. The HRs for overall mortality, disease-specific mortality, and progression were 1.30 (95% CI: 1.07–1.59). 1.3 (95% CI: 1.05–1.67), and 1.3 (95% CI: 1.07–1.59), respectively, in the CKD group from our database. The results of our multivariable analysis indicated that the preoperative eGFR value is a useful tool that can provide additional important information about patient prognosis when used in the context of the clinical and pathological outcomes.

For the past decade, increasing evidence has shown that renal insufficiency is frequent in patients with cancer and is related to increased cancer-specific mortality and overall mortality [22,23]. Previous studies have indicated a high prevalence of CKD in UTUC patients (40–59%) [13,14,15,24]. In accordance with previous studies, 41% of patients had preoperative CKD in this study. A high prevalence of CKD in UTUC and an association between preoperative renal insufficiency and poor oncological outcomes might result from the complexities of CKD-associated chronic infection/inflammation, oxidative stress, metabolic disorders, immunocompromised status, and frailty [25,26]. Those factors are not only related to reduced DNA repair capacity and lack of protection against viral oncogenes but are also associated with organ degradation and increased carcinogenicity [27,28,29]. Furthermore, many patients can be left with inadequate renal function and poor performance status after RNU to receive adjuvant chemotherapy, especially in patients with preoperative CKD status. Taken together, neoadjuvant chemotherapy can be better than adjuvant chemotherapy in patients with preoperative CKD status when the preoperative renal function is maintained to a certain extent. Additionally, kidney sparing surgery, such as endoscopic ablation or segmental ureterectomy, can be considered in patients with preoperative CKD because of clinical benefits in terms of renal function [30]. Kidney sparing surgery may protect from surgically induced renal insufficiency and could contribute to enhanced tolerance of adjuvant chemotherapy in appropriately selected patients with UTUC.

Our study has several recognized limitations worth mentioning. First and foremost, our data were derived from a multi-institutional database and retrospective analysis that included surgical, pathological, and follow-up differences among institutions. However, all surgeons and pathologists operated at institutions dedicated to management of UTUC. Second, information about socioeconomic status and comorbidities, such as smoking history and diagnosis of diabetes mellitus and cardiovascular disease, were not available in our database. This is a common methodological imitation when using multi-institutional retrospective databases. However, we tried to control for many potential confounding factors, including clinical and pathological characteristics, with multivariable statistical analyses. Third, we explored the predictive value of a single reduced preoperative eGFR value on oncological outcomes during follow-up after RNU. Chronic kidney disease is a general term used to describe heterogeneous disorders that affect the structure and function of the kidney [31]. A formal diagnosis of CKD in the preoperative setting can be challenging due to a lack of longitudinal data [32]. Furthermore, we did not assess the relationship between changes in postoperative eGFR and patient prognosis. A study conducted by Lane et al. [24] demonstrated CKD in 52% of patients with UTUC at diagnosis, with this rate significantly increasing to 78% after RNU, had a median relative reduction in renal function of 21%. However, in this study, we focused on the relationship between preoperative CKD status and prognosis.

Despite these limitations, this study demonstrated the negative impact of preoperative renal insufficiency on oncological outcomes in patients with UTUC who underwent RNU. An understanding of the association between preoperative renal function and oncological outcomes could offer better insights into the natural history of UTUC as well as improve patient counseling regarding risk assessment.

## 5. Conclusions

Our results indicate that preoperative renal insufficiency is a potential risk factor of poor oncological outcomes in patients with UTUC who underwent RNU. In our experience, preoperative renal insufficiency in patients who underwent RNU requires additional attention and strict follow-up strategies regarding patient progression and/or mortality.

## Figures and Tables

**Figure 1 biomedicines-10-01340-f001:**
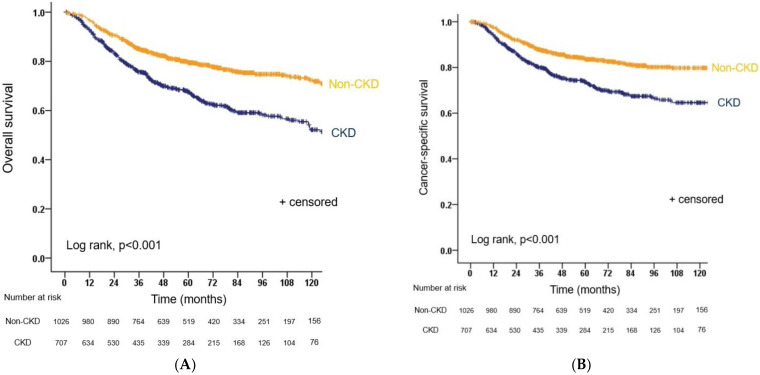
Kaplan–Meier analysis depicting: (**A**) overall survival; (**B**) cancer-specific survival; (**C**) progression-free survival; and (**D**) intravesical recurrence-free survival for patients with or without preoperative CKD. CKD, chronic kidney disease.

**Table 1 biomedicines-10-01340-t001:** Descriptive characteristics of 1733 patients treated with upper tract urothelial carcinoma.

	All Patients (n = 1733)	CKD (n = 707)	Non-CKD (n = 1026)	*p* Value
Age, years	65 (58–72)	69 (62–74)	63 (55–70)	<0.001
Male, n (%)	1276 (73.6)	499 (70.6)	777 (75.7)	0.017
BMI, kg/m^2^	24.2 (22.1–26.1)	24.3 (22.2–26.2)	24.2 (22.1–26.1)	0.521
Preoperative eGFR, mL/min/1.73 m^2^	64.4 (53.0–77.4)	50.1 (43.3–55.1)	74.9 (67.6–84.5)	<0.001
ASA score, n (%)				<0.001
1	428 (24.7)	120 (17.0)	308 (30.0)	
2	1101 (63.5)	470 (66.5)	631 (61.5)	
≥3	108 (6.2)	65 (9.2)	43 (4.2)	
Missing/unknown	96 (5.5)	52 (7.4)	44 (4.3)	
Previous bladder cancer, n (%)	219 (12.6)	99 (14.0)	120 (11.7)	0.155
Concomitant bladder cancer, n (%)	112 (6.5)	59 (8.3)	53 (5.2)	0.008
Surgical approach, n (%)				0.004
Open	1030 (59.4)	449 (63.5)	581 (56.6)	
Laparoscopy or robot	703 (40.6)	258 (36.5)	445 (43.4)	
Tumor laterality, n (%)				0.744
Right	795 (45.9)	321 (45.4)	474 (46.2)	
Left	938 (54.1)	386 (54.6)	552 (53.8)	
Tumor location, n (%)				<0.001
Renal pelvis	762 (44.0)	234 (33.1)	528 (51.5)	
Ureter	653 (37.7)	299 (42.3)	354 (34.5)	
Both renal pelvis and ureter	318 (18.3)	174 (24.6)	144 (14.0)	
Pathological T stage, n (%)				<0.001
pTis/pTa	241 (13.9)	71 (10.0)	170 (16.6)	
pT1	461 (26.6)	153 (21.6)	308 (30.0)	
pT2	319 (18.4)	144 (20.4)	175 (17.1)	
pT3	668 (38.5)	313 (44.3)	355 (34.6)	
pT4	44 (2.5)	26 (3.7)	18 (1.8)	
Tumor grade, n (%)				<0.001
Low grade	526 (30.4)	159 (22.5)	367 (35.8)	
High grade	1161 (67.0)	531 (75.1)	630 (61.4)	
Missing/unknown	46 (2.7)	17 (2.4)	29 (2.8)	
Concomitant LVI, n (%)	398 (23.0)	194 (27.4)	204 (19.9)	<0.001
Concomitant CIS, n (%)	248 (14.3)	108 (15.3)	140 (13.6)	0.341
Pathological N stage, n (%)				0.344
pNx	963 (55.6)	394 (55.7)	569 (55.5)	
pN0	648 (37.4)	250 (35.4)	398 (38.8)	
≥pN1	122 (7.0)	63 (8.9)	59 (5.8)	
Adjuvant chemotherapy, n (%)	376 (21.7)	188 (26.6)	188 (18.3)	<0.001

Continuous and non-normally distributed variables are presented as medians with interquartile ranges. CKD = chronic kidney disease; BMI = body mass index; eGFR = estimated glomerular filtration rate; ASA = American Society of Anesthesiologists; LVI = lymphovascular invasion; CIS = carcinoma in situ.

**Table 2 biomedicines-10-01340-t002:** Multivariable Cox proportional hazard regression analyses to predict death from all-cause, death from upper tract urothelial carcinoma, progression, and intravesical tumor recurrence in 1733 patients with upper tract urothelial carcinoma treated with radical nephroureterectomy.

Characteristics	OS			CSS			PFS			IVRFS		
	HR	95% CI	*p* Value	HR	95% CI	*p* Value	HR	95% CI	*p* Value	HR	95% CI	*p* Value
Age (continuous)	1.04	1.03–1.05	<0.001	1.03	1.01–1.04	<0.001	1.01	1.00–1.02	0.025	1.00	0.99–1.01	0.715
Sex												
Male	Reference			Reference			Reference			Reference		
Female	0.99	0.80–1.23	0.923	1.11	0.87–1.43	0.389	1.17	0.95–1.44	0.131	0.88	0.73–1.06	0.183
Body mass index (continuous)	0.97	0.94–1.00	0.035	0.99	0.95–1.02	0.396	0.98	0.95–1.01	0.120	0.98	0.96–1.00	0.077
ASA												
1	Reference			Reference			Reference			Reference		
2	0.97	0.76–1.23	0.784	0.95	0.73–1.24	0.713	0.83	0.66–1.03	0.087	1.04	0.86–1.25	0.702
≥3	1.20	0.81–1.78	0.361	1.06	0.66–1.71	0.798	0.97	0.65–1.43	0.865	1.14	0.81–1.61	0.455
Preoperative renal function												
Non-CKD	Reference			Reference			Reference			Reference		
CKD	1.30	1.07–1.59	0.010	1.33	1.05–1.67	0.016	1.31	1.07–1.59	0.008	1.18	1.00–1.39	0.056
Surgical approach												
Open	Reference			Reference			Reference			Reference		
Laparoscopy or robot	0.92	0.75–1.14	0.458	0.90	0.71–1.14	0.387	0.96	0.79–1.17	0.683	0.83	0.71–0.98	0.025
Previous bladder cancer												
No	Reference			Reference			Reference			Reference		
Yes	1.31	0.98–1.75	0.067	1.63	1.19–2.24	0.002	1.52	1.17–1.97	0.002	1.41	1.12–1.77	0.004
Concomitant bladder cancer												
No	Reference			Reference			Reference			Reference		
Yes	1.63	1.16–2.28	0.004	1.78	1.22–2.60	0.003	1.57	1.11–2.22	0.010	1.80	1.36–2.38	<0.001
Tumor location												
Renal pelvis	Reference			Reference			Reference			Reference		
Ureter	1.10	0.88–1.36	0.406	1.10	0.86–1.42	0.444	1.11	0.90–1.38	0.324	0.97	0.82–1.16	0.741
Both renal pelvis and ureter	1.12	0.86–1.46	0.413	1.16	0.86–1.57	0.323	1.21	0.94–1.57	0.143	1.12	0.90–1.41	0.313
Pathological T stage												
pTis/pTa/pT1/pT2	Reference			Reference			Reference			Reference		
pT3/pT4	2.46	1.94–3.11	<0.001	3.34	2.51–4.44	<0.001	2.74	2.17–3.46	<0.001	0.90	0.74–1.11	0.322
Tumor grade												
Low	Reference			Reference			Reference			Reference		
High	1.63	1.24–2.14	<0.001	1.87	1.32–2.66	<0.001	2.28	1.68–3.09	<0.001	0.98	0.82–1.18	0.850
Pathological N stage												
pNx	Reference			Reference			Reference			Reference		
pN0	1.27	1.03–1.57	0.028	1.02	0.79–1.30	0.903	0.83	0.67–1.02	0.082	0.98	0.84–1.16	0.847
≥pN1	2.27	1.67–3.08	<0.001	2.05	1.47–2.85	<0.001	1.84	1.39–2.44	<0.001	0.70	0.47–1.05	0.081
Concomitant LVI												
No	Reference			Reference			Reference			Reference		
Yes	1.84	1.48–2.28	<0.001	1.87	1.47–2.39	<0.001	1.81	1.47–2.23	<0.001	0.93	0.74–1.15	0.490
Concomitant CIS												
No	Reference			Reference			Reference			Reference		
Yes	1.05	0.81–1.36	0.721	1.13	0.85–1.51	0.403	1.06	0.83–1.35	0.643	1.13	0.91–1.42	0.278
Adjuvant chemotherapy												
No	Reference			Reference			Reference			Reference		
Yes	1.10	0.87–1.38	0.451	1.10	0.85–1.42	0.466	1.03	0.83–1.29	0.775	0.83	0.65–1.05	0.122

OS = overall survival; CSS = cancer-specific survival; PFS = progression-free survival; IVRFS = Intravesical recurrence-free survival; HR = hazard ratio; CI = confidence interval; ASA = American Society of Anesthesiologists; CKD = chronic kidney disease; LVI = lymphovascular invasion; CIS = carcinoma in situ.

## Data Availability

The data presented in this study are available on request from the corresponding author.

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
