# Peer review of "Clinical Implication of Preoperative Renal Function on Oncological Outcomes in Patients with Upper Tract Urothelial Carcinoma after Radical Nephroureterectomy"

_biomedicines, 2022, doi:10.3390/biomedicines10061340_

Round 1

Reviewer 1 Report

The authors present a multi-center retrospective study on the impact of preoperative chronic kidney disease (CKD) on oncological outcomes in patients with upper tract urothelial carcinoma (UTUC) submitted to radical nephroureterectomy (RNU).

The main findings were that overall (OS), cancer-specific (CSS) and progression-free survival (PFS) were lower in patients with preoperative CKD (defined as eGFR < 60).

The manuscript is clearly written and well organized. The authors identified important limitations in their study.

Some questions need clarification or correction:

Materials and Methods

Please specify which were the criteria for adjuvant chemotherapy. Even in the non-CKD grouo, only a small proportion of patients eventually eligible for adjuvant chemotherapy (>pT1 or pN+) received it.

Results

Line 164 – the 5-year IVRFS of 56.4% for CKD and 54.2% for non-CKD is not concordant with figure 1D (it seems to be the opposite).

Line 177 – the lower limit of hazard ratio interval is 1.07 (and not 1.70).

Discussion

Line 216 – the lower limit of hazard ratio interval is 1.07 (and not 1.70).

Since patients with pre-operative CKD have worse prognosis, it would be worth discussing the role of neoadjuvant treatment in these patients, as often RNU renders them ineligible to post-op treatment.

Reviewer 2 Report

Upper tract urothelial carcinoma (UTUC) represents the 5-10% of urothelial carcinomas overall. The gold standard treatment is still represented by radical nephroureterectomy (RNU) with bladder cuff excision. However, the prognosis of UTUC patients remains poor, with relatively high recurrence rates >30% at five years and estimates of cancer-specific survival ranging between 60-80% at five years. Chronic kidney disease (CKD) has long been recognized as an independent predictor of adverse postoperative outcomes including cardiopulmonary complications, prolonged length of hospital stay, and increased mortality for patients undergoing vascular, general, and urological oncological surgery. As a result, CKD could impact negatively on oncological outcome in UTUC patients. The aim of the study was to evaluate the impact of preoperative CKD on oncological outcomes in UTUC patients underwent to RNU.

Authors should be congratulated for their work. The manuscript is well-written. Despite the dedication, the topic is already discussed by several previous authors (DOI: 10.1111/bju.12597 ; DOI: 10.1016/j.euf.2018.03.003. Moreover, several points warrant a mention:

  1. Are data available on concomitant therapies of UTUC patients could impair renal function? Which were the comorbidities of these patients?
  2. How did Authors define the diagnosis of CKD?
  3. Are data available on presence of hydronephrosis in UTUC patients underwent RNU?
  4. When lymph nodes dissection was performed, how many lymph nodes were dissected? Are data available? Among that patients’ subgroup, how did the preoperative CKD impact on the oncological outcomes?
  5. Which was the protocol of adjuvant chemotherapy adopted? In 188 CKD patients underwent to adjuvant chemotherapy how was the eGRF stratified? Did all patients receive the cisplatin?
  6. Clinically, which were the implication of CKD diagnosis on UTUC patients? How would change the management of these patients? Authors should better clarify this aspect.

Reviewer 3 Report

The efforts of the authors are praiseworthy, the manuscript is well written, references are accurate. However, the manuscript shows several meaningful flaws that should be better addressed:

MAJOR ISSUES

Major limitation of the present manuscript concerns the results statistical evaluation that should be better addressed. Due to the complexity of different clinical and oncologic features burden on OS, CSS and PFS, the lack of potential confounders assessment such as diabetes, cardiovascular diseases and smoking status may lower the reliability of reported results.

As regards oncologic features listed, several diriment clinical and histological elements are missing. In order to properly characterize the study population and to support evidences clinical value, histological tumor nature should be assessed. Moreover, clinical T and N stage at diagnosis should be evaluated and related with kind of surgery performed as well as recurrence free survival. In this light also information regarding the recurrence location and the metastatic disease clinical and surgical management should be integrated within Table 1.

Although renal function indicators are recorded in pre-treatment phase, the lack of assessment of early postoperative and long-term AKI should limit the reliability of reported results.  Moreover, the assessment of tumor location should be implemented within table 1.

MINOR ISSUES

The definition of High volume centre should be added in the materials and methods section.

Author Response

Response to Reviewer 3 Comments

The efforts of the authors are praiseworthy, the manuscript is well written, references are accurate. However, the manuscript shows several meaningful flaws that should be better addressed:

MAJOR ISSUES

1. Major limitation of the present manuscript concerns the results statistical evaluation that should be better addressed. Due to the complexity of different clinical and oncologic features burden on OS, CSS and PFS, the lack of potential confounders assessment such as diabetes, cardiovascular diseases and smoking status may lower the reliability of reported results.

Response: Thank you for your comment. We absolutely agree your opinion. As we mentioned in limitations, information about socioeconomic status and comorbidities that could affect renal impairment were not available in our database.

2. As regards oncologic features listed, several diriment clinical and histological elements are missing. In order to properly characterize the study population and to support evidences clinical value, histological tumor nature should be assessed. Moreover, clinical T and N stage at diagnosis should be evaluated and related with kind of surgery performed as well as recurrence free survival. In this light also information regarding the recurrence location and the metastatic disease clinical and surgical management should be integrated within Table 1.

Response: Histologic tumor natures, such as pathological T stage, pathological N stage, tumor grade, and the presence of lymphovascular invasion or carcinoma in situ, were already presented in Table 1 of our manuscript. We thought that pathological stage was more important than clinical stage in assessing characteristics of the study papulation and evaluating the impact of the oncological outcome. We are sorry but we cannot assess the information regarding the recurrence location and following management due to multicenter and retrospective nature of this study. 

3. Although renal function indicators are recorded in pre-treatment phase, the lack of assessment of early postoperative and long-term AKI should limit the reliability of reported results. Moreover, the assessment of tumor location should be implemented within table 1.

Response: We agree with your opinion. As we mentioned in limitations, a formal diagnosis of CKD in the preoperative setting can be challenging due to a lack of longitudinal data. Data about tumor location was already presented in Table 1 of our manuscript classified as renal pelvis, ureter, and both renal pelvis and ureter.

MINOR ISSUES

1. The definition of High volume centre should be added in the materials and methods section.

Response: Thank you for your opinion. However, we did not use the terminology ‘High volume centre’ in our manuscript.

Reviewer 4 Report

Dear Editor, thank you so much for inviting me to revise this manuscript about UTUC.

This study addresses a current topic.

The manuscript is quite well written and organized. English could be improved.

Figures and tables are comprehensive and clear.

The introduction explains in a clear and coherent manner the background of this study.

We suggest the following modifications:

  • Introduction section: although the authors correctly included important papers in this setting, we believe a couple of studies should be cited within the introduction ( PMID: 34387596; PMID: 32798146), only for a matter of consistency. We think it might be useful to introduce the topic of this interesting study.
  • Methods and Statistical Analysis: nothing to add.
  • Discussion section: Very interesting and timely discussion. Of note, the authors should expand the Discussion section, including a more personal perspective to reflect on. For example, they could answer the following questions – in order to facilitate the understanding of this complex topic to readers: what potential does this study hold? What are the knowledge gaps and how do researchers tackle them? How do you see this area unfolding in the next 5 years? We think it would be extremely interesting for the readers.

However, we think the authors should be acknowledged for their work. In fact, they correctly addressed an important topic, the methods sound good and their discussion is well balanced.

One additional little flaw: the authors could better explain the limitations of their work, in the last part of the Discussion.

We believe this article is suitable for publication in the journal although major revisions are needed. The main strengths of this paper are that it addresses an interesting and very timely question and provides a clear answer, with some limitations.

We suggest a linguistic revision and the addition of some references for a matter of consistency. Moreover, the authors should better clarify some points.

Author Response

Response to Reviewer 4 Comments

Dear Editor, thank you so much for inviting me to revise this manuscript about UTUC.

This study addresses a current topic.

The manuscript is quite well written and organized. English could be improved.

Figures and tables are comprehensive and clear.

The introduction explains in a clear and coherent manner the background of this study.

We suggest the following modifications:

  • Introduction section: although the authors correctly included important papers in this setting, we believe a couple of studies should be cited within the introduction ( PMID: 34387596; PMID: 32798146), only for a matter of consistency. We think it might be useful to introduce the topic of this interesting study.

Response: Thank you for your recommendation. As your recommendation, we added some sentences regarding the perioperative treatment as follows. “Considering the poor prognosis of UTUCs treated with RNU, the addition of perioperative chemotherapy can be expected to have a synergistic effect to improve patients’ outcomes (PMID: 34387596; PMID: 32798146). Several prognostic factors have been investigated to allow clinicians to risk stratify and better select patients for perioperative chemotherapy, either in the neoadjuvant or adjuvant setting.” These sentences are now presented on Introduction (Page 2, line 57) of the revised manuscript.

  • Methods and Statistical Analysis: nothing to add.
  • Discussion section: Very interesting and timely Of note, the authors should expand the Discussion section, including a more personal perspective to reflect on. For example, they could answer the following questions – in order to facilitate the understanding of this complex topic to readers: what potential does this study hold? What are the knowledge gaps and how do researchers tackle them? How do you see this area unfolding in the next 5 years? We think it would be extremely interesting for the readers.

However, we think the authors should be acknowledged for their work. In fact, they correctly addressed an important topic, the methods sound good and their discussion is well balanced.

Response: Thank you for your comment. We added some sentences to expand the Discussion section.

One additional little flaw: the authors could better explain the limitations of their work, in the last part of the Discussion.

Response: Thank you for your comment. However, we think that we were mentioning how we dealt with potential limitations in Discussion session.

We believe this article is suitable for publication in the journal although major revisions are needed. The main strengths of this paper are that it addresses an interesting and very timely question and provides a clear answer, with some limitations.

We suggest a linguistic revision and the addition of some references for a matter of consistency. Moreover, the authors should better clarify some points.

Response: Thank you for your comment. Our manuscript has been thoroughly revised regarding grammar and copy editing by the professional English editing company (E-World editing).

Round 2

Reviewer 2 Report

congratulations for the revised manuscript.

the points raised have been adequately addressed.

Reviewer 4 Report

Acceptance.